



# Water vapor density and turbulent fluxes from three generations of infrared gas analyzers

Seth Kutikoff[1], Xiaomao Lin[1], Steven R. Evett[2], Prasanna Gowda[3], David Brauer[2], Jed Moorhead[2], Gary Marek[2], Paul Colaizzi[2], Robert Aiken[1], Liukang Xu[4], and Clenton Owensby[1]

[1]Department of Agronomy, Kansas State University, Throckmorton Plant Sciences Center, Manhattan, KS, 66506, USA

[2]USDA-ARS Conservation & Production Research Lab, 300 Simmons Road, Unit 10, Bushland, TX, 79012, USA

[3]USDA-ARS 141 Experiment Station Road, Stoneville, MS, 38776, USA

[4]LI-COR Bioscience, 4647 Superior Street, Lincoln, NE, 68504, USA

*Correspondence to*: Xiaomao Lin (xlin@ksu.edu)

**Abstract.** Fast-response infrared gas analyzers (IRGAs) have been widely used over three decades in many ecosystems for long-term monitoring of water vapor fluxes in the surface layer of the atmosphere. While some of the early IRGA sensors are still used in these national and/or regional eco-flux networks, optically-improved IRGA sensors are newly employed in the same networks. The purpose of this study was to evaluate the performance of water vapor density and flux data from three generations of IRGAs – LI-7500, LI-7500A, and LI-7500RS (LI-COR Bioscience, Inc., Nebraska, USA) – over the course of a growing season in Bushland, Texas, USA in an irrigated maize canopy for 90 days. The energy balance ratio, which is the sum of turbulent fluxes divided by the sum of surface available energy, was used to assess systematic biases of the IRGA sensors for evapotranspiration (ET). Water vapor density measurements were in generally good agreement, but temporal drift occurred in different directions and magnitudes. Means exhibited mostly shift changes that did not impact the flux magnitudes, while variances of water vapor density fluctuations were occasionally in poor agreement, especially following rainfall events. LI-7500 variances were largest compared to recent LI-7500RS and LI-7500A results manifesting in widened cospectra, especially under unstable and neutral static stability. Agreement among the sensors was best under the typical irrigation-cooled boundary layer, with a 14% interinstrument coefficient of variability under advective conditions. Generally, the smallest variances occurred with the LI-7500RS, and high-frequency spectral corrections were larger for these measurements resulting in similar fluxes between the LI-7500A and LI-7500RS. Fluxes from the LI-7500 were best representative of growing season ET based on a world-class lysimeter reference measurement but using the energy balance ratio as an estimate of systematic bias corrected most of the differences among measured fluxes.

## 1 Introduction

The eddy covariance (EC) method is a standard way to monitor water vapor flux between the surface and atmosphere at most spatial scales and environments, including marine (Honkanen et al., 2018; Takahashi et al., 2005), forest (Novick et al., 2013; Zhang et al., 2012), grassland (Haslwanter et al., 2009; Hirschi et al., 2017), and cropland (Ding et al., 2013; Kochendorfer



and Paw, 2011). In water-limited regions, the need to conserve a subsurface source, such as the U.S. Ogallala Aquifer, serves as motivation for agricultural producers to estimate the crop water use for daily irrigation scheduling (Xue et al., 2017). Current crop production involves innovative water saving measures, such as variable rate irrigation management, requiring high quality

evapotranspiration (ET) data to supplement efforts to calculate the correct amount of water to apply to crops (O'Shaughnessy et al., 2016). In ecosystem networks both large (FLUXNET Baldocchi et al., 2001), and small (e.g., Delta-Flux see Runkle et al., 2017), as well as at individual research fields in Texas (Evett et al., 2012a) and California (Oncley et al., 2007), the IRGAs built by LI-COR Biosciences, Inc. (Lincoln, Nebraska, USA) have been widely used for over two decades to monitor water vapor fluxes. The accuracy of ET measurements relative to a reference system can be assessed to investigate potential

systematic problems with instrumentation (Mauder et al., 2006). Based on this analysis, an open-path, nondispersive infrared gas analyzer (IRGA) has long been selected as the standard fast-response hygrometer for decades after the era of Lyman-alpha and krypton hygrometer absorption sensors (absorption of ultraviolet radiation by water vapor, e.g., Kaimal and Finnigan, 1994). The optical sensor of the IRGA detects water vapor through differential or ratio measurement of infrared transmittance at two adjacent wavelengths with one located in a region of large water vapor absorption and the other where absorption is

negligible (Kaimal and Finnigan, 1994). The transmitting path is typically 0.2-1.0 m long, and beams are usually modulated by a mechanical chopper to permit high-gain amplification of the detected signal. Generally, such an optical approaching device is unreliable when air humidity reaches saturation (rainfall or dew) because of liquid water present in optical pathways. The ratio detecting technique used to improve the signal-noise ratio of water vapor signals and also removes the common noise in the absorption path length. For water vapor detected in all LI-COR 7500 models, the ratio of these measurements determines

an estimate of vapor absorptance, which is converted to a concentration or density (absolute humidity), $\rho_v$, using a third-order calibration polynomial. Any biases occurring in this absorption, therefore, propagate to $\rho_v$ measurement errors. Fratini et al. (2014) described contributing factors to this error, including the magnitude of absorptance fluctuations, and showed that zero drift in $\rho_v$, or in other words, the change in bias over time, tends to occur in steps rather than in a continuous fashion.

The IRGA specifications for water vapor density measurement $\rho_{v,m}$, including accuracy, precision, and drift have been

unchanged over three models of sensors: LI-7500, LI-7500A, and LI-7500RS. The LI-7500 was first introduced in 1999, followed by the LI-7500A in 2010 and LI-7500RS in 2016. The differences between the LI-7500 and LI-7500A reported by LI-COR primarily address electrical power requirements in cold climate conditions and ease of use. Progressing from the LI-7500A to the LI-7500RS, while no physical differences are evident, optical changes were made to improve the stability of measurements in the presence of window contamination which can cause systematic bias (Heusinkveld et al., 2008). LI-COR

reported that $\rho_{v,m}$ drift was more than an order of magnitude smaller in the LI-7500RS than the original LI7500A and was accompanied by reduced interinstrument variability (Burba et al., 2018). They also found that after rainfall, LI-7500A and LI-7500RS measurements were similar but agreement lessened after approximately one week. As the duration of IRGA deployment increases from weeks to months and years, calibration becomes more important to ensure accuracy for fast-response water vapor measurements since their measurement stability is relatively low (Iwata et al., 2012). The factory

calibration procedure, resulting in span and zero coefficients, consists of measured water vapor density being compared to the





absorption of water vapor from a dewpoint generator over a range of temperatures from 17 to 41°C. Based on the manufacturer calibration and re-calibration sheets (after a certain period the IRGA is returned to the manufacturer for re-calibration), the span drift is primarily a function of temperature, whereas the zero drift is chiefly influenced by the measurement range of water vapor density.

In addition to the IRGA, a sonic anemometer is necessary to determine water vapor flux. This pair of instruments introduces systematic error due to their physical separation, which is a source of high frequency turbulent signal loss (Massman, 2000). The magnitude of flux attenuation is enhanced by lighter wind speed and a greater ratio of horizontal separation to sensing height (Horst and Lenschow, 2009). The expected cospectra, or eddy flux in the spectral domain, can be estimated analytically with a series of transfer functions (Massman, 2000; Moncrieff et al., 1997) that account for signal loss

at low and high frequencies. A spectral correction factor can often be determined based on how this modeled cospectrum departs from the measured cospectrum, indicating the degree of flux loss for a given observation period and EC system.

To address offset errors of water vapor density from an IRGA, data are typically compared to another type of sensor. In a comparison to the enclosed-path EC155 system (Campbell Scientific, Logan, UT, USA), errors in water vapor density were generally between -3 and 3 g m$^{-3}$ (Novick et al., 2013). Such errors were largest in early to mid-morning hours coinciding with

the likely formation of dew and fog, and after bias correction, the linear regression slope and offset were 1.01 and 1.68 g m$^{-3}$, respectively. In a study involving an LI-7500 and Krypton hygrometer in a semi-arid climate where rainfall is irregular (34.6 mm in three events from approximately three weeks of data), flux comparisons were made using simple linear regression (Martínez-Cob and Suvočarev, 2015). With the Krypton hygrometer being unable to measure absolute concentration of water vapor, comparisons of $\rho_v$ data were not made. In this case, $\rho_v$ can be calibrated to a sensor explicitly designed to determine

absolute humidity. This calibration should be stable (avoid short timescale error) and not drift (avoid long timescale error). In an environment prone to contamination, the measurement timeframe could be 1–2 weeks (Iwata et al., 2012). Accurate water vapor determination is also crucial in flux processing procedures, specifically to account for air density fluctuations which complicate the effect of error propagation into water vapor flux (Fratini et al., 2014).

Due to the high expense of infrared gas analyzers (IRGA), there is little research intercomparing multiple instruments

except by the manufacturer itself. Historically, instrumentation errors from EC systems average 10–20%, with additional contributions from random errors and a smaller, non-negligible amount from systematic bias (Alfieri et al., 2011). Gas analyzers from the same manufacturer have been shown to differ in short-term drifts (Moncrieff et al., 2004). Here, we assess three generations of LI-7500 instruments in advective field conditions over 90 days by evaluating differences in water vapor density measurements and how those differences impact the estimation of the turbulent exchange of water vapor compared

with that measured using a large weighing lysimeter. Flux characteristics and how they deviate over the course of the growing season are also analyzed to determine any advantages a newer water vapor analyzer may have over earlier models.



## 2 Data and Methods

### 2.1 Site description and measurements

The field study was conducted between 16 June [day of the year (DOY) 168] and 13 September 2016 (DOY 257) on the
lysimeter field at the USDA-ARS Conservation & Production Research Laboratory, Bushland, Texas, located in the Texas
panhandle (35.19° N, 102.09° W, 1170 m elevation above sea level). Corn (*Zea mays* L.) was planted on 10 May, with
emergence eleven days later, and thereafter crop height grew steadily during the first part of the study period. From 20 June to
19 July, crop height $h_c$ increased nearly linearly from 0.85 m to its peak of 2.30 m. After this point, plants were in their
reproductive stage with a decreasing leaf area index trend ensuing. The high ET demand of corn during its development is well
known and necessitated irrigation to complement precipitation. Both in intensity and frequency, precipitation was erratic (Evett
et al., 2019) as typical for a semi-arid climate, which is mostly in the range of 250–350 mm (Gowda et al., 2009; Tolk et al.,
2013) during the corn growing season at Bushland.

   The EC experiment included three systems consisting of IRGA models LI-7500, LI-7500A, and LI-7500RS, with a sonic
anemometer (CSAT3, Campbell Scientific, Inc., Logan, UT, USA), sampling at 20 Hz. Each IRGA outputs $CO_2$, $H_2O$,
barometric pressure, and a diagnostic value indicating signal strength and statuses of optical wheel rotation rate, detector
temperature, and chopper temperature. The gas analyzers were mounted at a height of 4.6 m above the ground ($\geq 2\ h_c$), facing
southward with the anemometers situated west of the gas analyzers perpendicular to the dominant (southerly) wind direction.
Two systems (EC1 and EC2) were at a tower instrumented with an LI-7500RS, LI-7500A, and CSAT3. The horizontal
separation between gas analyzer and sonic anemometer was approximately 10 cm and 20 cm, respectively. This spacing on
the same tower is comparable to a recent intercomparison of fluxes from two open-path IRGAs (Polonik et al., 2019). The
third system (EC3) affixed on a tower 26 m to the south, had an LI-7500 and CSAT3 separated by 10 cm horizontally. All gas
analyzers were approximately 10 cm lower than the sonic anemometers and angled slightly downward in accordance with the
manufacturer's recommendation to reduce collection of water droplets and contamination on the lens. Both towers had
reference $\rho_v$ data from an air temperature-humidity probe (HMP 155A, Vaisala, Helsinki, Finland) containing a capacitive-
type humidity sensor (HUMICAP 180R, Vaisala, Helsinki, Finland). Ancillary data were taken of net radiation $R_n$ (NR-LITE2,
Kipp & Zonen, Delft, The Netherlands) at 2.6 m above ground, soil heat flux $G$ (HFT-3.1, Radiation and Energy Balance
Systems, Seattle, WA, USA) at 8 cm below ground, and thermistors and water-content reflectometers (CS655, Campbell
Scientific, Inc., Logan, Utah, USA) at 2 and 6 cm below ground, which were used to estimate soil heat storage (Kutikoff et al.,
2019).

### 2.2 Data processing and statistical analysis

Water vapor density data among the three infrared open–path IRGAs were compared in a fashion similar to Mauder et al.
(2006). The following characteristics of variance ($\overline{\rho_v'\rho_v'}$) and covariance ($\overline{w'\rho_v'}$) were of interest: regression intercept (*a*),





slope (*b*), and coefficient of determination ($r^2$); root mean square deviation (*rmsd*); and bias (*d*). Comparability between LI-7500RS and the other two models was found using *rmsd*, defined as:

$$rmsd = \sqrt{\sum(x_{A,i} - x_{RS,i})^2},$$
(1)

where $x_{A,i}$ is the $i^{th}$ observation for the LI-7500/A and $x_{RS,i}$ is the $i^{th}$ observation for the LI-7500RS. Interinstrument variability was also determined, which is like *rmsd* except the mean of the EC systems is the reference value. For fluxes, interinstrument variability was expressed relative to flux magnitude using the coefficient of variation ($CV_{I-I}$). Implausible values of 20 Hz data, defined as greater than 30 g m$^{-3}$ or less than 2 g m$^{-3}$, were removed prior to taking half-hourly means. Additionally, while the

LI-7500A and LI-7500 were calibrated in 2014 and 2015, a correction to these data was made based on a factory calibration after data was collected. Otherwise, no additional conditioning was performed on the raw data. Given the interest in sensor sensitivity, comparisons were also made between collocated HMP155A and IRGA(s) at each tower, which were assumed to be sensing identical air parcels containing equal water vapor density.

To further ascertain the performance of IRGAs, (co)spectral density of $\rho_v$ ($w\rho_v$) measurements were calculated for each

of three EC systems using Welch's periodogram method (Blanken et al., 2003). The distribution of power across frequencies, particularly signal loss at high frequencies, can indicate differences in flux characteristics with an expectation that latent heat would be underestimated. Of particular interest are results from an advective environment in which high frequency variation is enhanced (Prueger et al., 2012). Data were conditioned by linear detrending on half-hour (36,000 points) segments (Zhang et al., 2010). Spectral density ($S_{\rho_v}$) was calculated across these segments with a Hamming window length of 360 and overlap

of 180 observations. Then the spectra were averaged into 100 evenly spaced bins on the logarithmic scale. The same procedure was repeated for the cospectra of vertical velocity and water vapor density, indicating the behavior of water vapor flux in the spectral domain. Finally, ogives were calculated to summarize differences in cospectra across wavelengths by integrating the cospectra from low-frequency energy to high-frequency energy on a scale from 0 to 1. The (co)spectra and ogives were multiplied by the frequency and normalized by mean (co)variance to make the data dimensionless.

After examining raw variances and covariances, water vapor fluxes (*E*) were processed using Eddypro (v6.2.0) software (LI-COR Bioscience, Lincoln, Nebraska, USA) for half-hour averaging periods when availability of data exceeded 90% (*w* and $\rho_v$ were recorded for at least 32,400 of 36,000 possible observations). Prior to computing fluxes, a statistical screening of time series data was implemented. Spikes were detected using the median absolute deviation for each half-hour (Mauder et al., 2013) and replaced with the half-hour mean of non-outlier observations. Then data was detrended by block average and

corrections were made to account for sensor separation, tilt of the sonic anemometer via double rotations (Fratini and Mauder, 2014), and spectral energy loss in both low (Moncrieff et al., 2004) and high (Moncrieff et al., 1997) frequency ranges. Based on spectral losses and other corrections, *E* was calculated iteratively. The original water vapor flux was multiplied by the spectral correction factor of $\overline{w'\rho_v'}$ before adding WPL density fluctuation terms (Kaimal and Finnigan, 1994). Sensible heat (*H*) was then corrected for humidity effects that arise from using sonic temperature in place of air temperature (Van Dijk et

al., 2004). Finally, this corrected *H* was multiplied by its spectral correction factor, and the WPL term was added to the



corrected water vapor flux to create a final $E$ or $\lambda E$. Approximately 13.5% of available data were removed through results of steady-state and fully developed turbulence tests (Mauder and Foken, 2004). The acquisition ratio of each half-hour was obtained by dividing the count of non-filtered fluxes by the maximum number of observations (Kim et al., 2015).

Intercomparison of $\lambda E$ and its systematic error (δ) and random uncertainty (ε) components was conducted on half-hourly and daily timescales. The measured $\lambda E$ is assumed to be the difference between the actual flux and these errors (Lasslop et al., 2008). Systematic error can be evaluated in the context of surface energy balance, such that δ is zero when turbulent flux equals the available energy measured through solar radiation, ground heat flux, and heat storage during a given period (Mauder et al., 2013). The estimate of systematic error is then

$$\delta = \lambda E(\frac{1}{EBR} - 1), \quad (2)$$

and

$$EBR = \frac{H+\lambda E}{R_n - G - J}, \quad (3)$$

where the terms in the numerator are independent (H is sensible heat flux, and $\lambda E$ is latent heat flux) for each EC system and those in the denominator are shared among the EC systems. $J$ was calculated as the sum of soil and photosynthesis heat storage since the other components of heat storage contribute negligibly to instantaneous energy balance in this ecosystem (Kutikoff

et al., 2019). Random error associated with sampling was quantified with the method of Finkelstein and Sims (2001), which calculates the variance of the covariance using the raw timeseries data for each averaging period. Together, error quantification can indicate if half-hour fluxes from the three EC systems statistically differ for half-hours in which turbulent flux measurements are reliable.

Water vapor flux was compared using the equivalent total water depth $ET$ for daily totals. Gap filling, following Reichstein
et al. (2005), was done for half-hours that were flagged for any of the three EC systems based on steady–state and developed turbulence tests (Mauder and Foken, 2004), occurrence of precipitation, and high relative humidity (RH > 95%). Total gap-filled ET was close to the sum of the half-hour observations, with approximately a 3% greater flux for each EC system. Flux accuracy of the three EC systems was assessed in relation to a large weighing lysimeter, which has an accuracy of 0.05 mm hr[-1] (Evett et al., 2012b). Located within 30 m of the EC system, lysimeter ET was computed using a soil water balance
approach from a subsection of the same field. Briefly, the mass change of water measured by the weighing lysimeter was calculated and converted into a flux based on the surface area of the lysimeter and density of water. Description of the lysimeter data can be found in Moorhead et al. (2017).

## 3 Results

The findings of the study are presented in three subsections, including water vapor density mean and fluctuations, spectra and
cospectra, and fluxes. All were influenced by irrigation and precipitation events. Water added to the field included 498 mm from 33 separate subsurface drip irrigations (SDI) (Evett et al., 2019) and 238 mm of precipitation (Evett et al., 2018),



consistent with an average growing season (Gowda et al., 2009; Tolk et al., 2013). However, much of that rainfall (88%) occurred after 1 August, and combined with crop maturity, eliminated the need for irrigation after 18 August.

Data filtering also impacted all comparisons. After all threshold and precipitation screenings, 3,577 out of a possible 4,320 half-hour observations are available for analysis. The acquisition ratio was comparable to similar studies (Wu et al., 2015). Between 9:00 AM and 9:00 PM (LST), the ratio exceeded 92%, whereas EC system issues reduced availability in the predawn hours to as low as 61% for the half-hour ending at 7:00 AM (Fig. 1).

### 3.1 Water vapor density validation

The long-term zero drift of water vapor density for the three IRGAs was evaluated as the three-month change in bias $\Delta\rho_v$. As

the study period began, the reference value of water vapor density $\rho_{v,r}$ ranged from 3 to 18 g m$^{-3}$. Accordingly, the measured values $\rho_{v,m}$ for the LI-7500 and LI-7500RS were biased low and the LI-7500A was biased high. After applying the post-correction to the LI-7500 and LI-7500A data, all $\rho_{v,m}$ were between 0.11 and 1.31 less than $\rho_{v,t}$ (Fig. 2). At the end of the study period, all IRGAs clearly showed an increased bias relative to the HMP155. Interestingly, the LI-7500 and LI-7500A had moved towards larger values, whereas the LI-7500RS moved towards smaller values (Fig. 2). That resulted in the LI-7500

$\Delta\rho_v$ decreasing, whereas the other two newer analyzers ended with greater $\Delta\rho_v$. The magnitude of bias was larger for the LI-7500 and LI-7500A than the LI-7500RS and a similar degree of day/night variability (sensitivity to solar radiation) was apparent among the IRGAs regardless of $\overline{\rho_v}$. These temporal patterns may indicate a low frequency modulated signal hidden in the instruments.

The divergence of $\rho_{v,m}$ between early and late times in the study period is the result of many short-term changes in bias.

To assess short-term drift $\Delta\rho_v$, half-hour differences between LI-7500s and HMP155s were calculated, with each timeseries bias corrected to set the initial value to zero (Fig. 3). The magnitude of daily drift averaged 0.09 g m$^{-3}$ for the LI-7500RS, 0.1 g m$^{-3}$ for the LI-7500A, and 0.13 g m$^{-3}$ for the LI-7500. Over 10-day periods, this increased to 0.36, 0.27, and 0.29, respectively. Rainfall contributed to the bulk of changes in drift. Rain-free periods as noted over the initial 10 days, gave the best insight into the stability of the sensors, and suggested that the LI-7500RS performed best. However, the extended dry period between

DOY 186 and DOY 196 suggested the opposite, when the LI-7500RS suffered from large short-term drift. After this time, the LI-7500RS appeared to be more stable, with steady *rmsd* over the final 50 days compared to the other two instruments.

According to Figs. 2 and 3, analyzer performance differed between day and night. This diel cycle is indicative of a solar radiation-induced error (Mauder et al., 2006; Miloshevich et al., 2009) and although amplitude varies, it appeared most prominently for the LI-7500 and least substantially for the LI-7500A. Periods with more instrument drift were coincident

mainly with larger cycles, but the sudden performance change of the LI-7500RS on DOY 191 did not reflect this tendency. Accidental window contamination may explain this observation, with typical behavior of absolute humidity from the LI-7500RS resuming from DOY 192 onward.

To investigate the unexpected large drift exclusive to the LI-7500RS on DOY 191, biometeorological data were assessed. Light southerly winds and moderately humid conditions were observed when $\Delta\rho_{v,RS}$ increased from -0.96 to -2.45 between
8:30 and 9:30 PM LST. While nothing unusual occurred meteorologically, a 3°C drop in temperature and 10% increase in RH accompanying the loss of daytime heating was noted. It was instructive to look at the variation in RH as estimated using vapor and ambient pressure from the IRGAs and sonic temperature from the CSAT3. While the magnitude of RH did vary slightly among the sensors, the increase in RH was similar for the LI-7500 and LI-7500A while being less than half for the LI-7500RS. In the hours immediately prior and after, the slopes of $\Delta\rho_{v,RS}$ among the IRGAs and HMPs are nearly in lockstep. Unlike other
deviations that exist on a subdaily timescale, this new offset continued until DOY 197. Step changes are a dominant feature in the linear regression between $\rho_{v,75/A}$ and $\rho_{v,75RS}$.

Differences between the means and fluctuations of $\rho_v$ are summarized in Fig. 4 as a function of day of year. Since variance of the $\rho_v$ time series reflects the mean of squared fluctuations $\overline{\rho_v'^2}$, greater variance in the half-hourly data reflects larger fluctuations $\rho_v'$. While the LI-7500 tended to have consistently greater $\overline{\rho_v'^2}$ values, the comparison between the LI-7500A and
LI-7500RS was more complicated. For example, the LI-7500A initially had slightly larger or the same fluctuations as the LI-7500RS for most daytime observations. After noon on DOY 196, the LI-7500RS consistently began to have larger fluctuations. Then from midday on DOY 226 to DOY 232 noon, the pattern flipped again. Following DOY 232, agreement was consistently close until DOY 254, and greater fluctuations from the LI-7500A were again found through the remainder of the study period. Even when the LI-7500RS fluctuations tended to be relatively large, it did not have the large overestimation of fluctuations
observed periodically with the LI-7500A, such as noted on DOY 184, 190, 193, 211, 216, and 253. While the stochastic nature of turbulence is partially responsible for the large scatter in $\overline{\rho_v'^2}$ shown in Fig. 4, the degree of variance in the older sensors exceeded that of the LI-7500RS.

Agreement between $\overline{\rho_v}$ of the LI-7500RS and the older IRGAs was generally strong and stable despite occasional large errors. In the first week of the study, regardless of the absolute error, linear regression parameters indicated well-calibrated
measurements for the purpose of eddy covariance, in which offset has no effect on the statistic. During the middle 30 days of the study, agreement was also high, reflected by $r^2$ values of 0.94 and 0.97 and slopes of 0.98 and 0.93, respectively for LI-7500 and LI-7500A. Little change from those parameters occurred across a wide range of $\rho_v$ during the final 30 days of the study, when lower temperature and higher relative humidity reduced evaporative demand. As expected, greater comparability in $\overline{\rho_v}$ was accompanied by a small $\rho_v'$ error. However, while step changes in $\overline{\rho_v}$ occurred, $\rho_v'$ did not change over time.

Variance of water vapor density $\overline{\rho_v'^2}$ was compared using the LI-7500RS as reference, for the entire dataset including daytime and advective periods only (Table 1). Nighttime estimates were particularly prone to overestimation by the LI-7500. Advective periods were prone to greater errors while having reduced interinstrument variability.




## 3.2 Spectra and cospectra

Since the three analyzers had the same specifications and were configured to measure turbulence in the same fashion, any
deviations in spectral characteristics would be an indication of possible drift. Returning to the distinct LI-7500RS error on
DOY 191, spectra were examined during the interval from 8:00-9:30 PM (LST), which consisted of three spectra corresponding
to consecutive flux averaging periods. Overall, as evident from Fig. 5a–c, the shapes of spectra were in close agreement during
the daytime, whereas the nighttime peak frequency was shifted to lower frequencies indicating the predominance of large
eddies after sunset. At 8 PM, the three spectra were nearly identical and matched the predicted -2/3 slope (Fig. 5d). In the
following hour, the spectra of the LI-7500A and LI-7500 remained nearly identical, whereas the LI-7500RS spectra were
greatly modified. Based on the 20 Hz timeseries, air humidity began to decrease suddenly at roughly 8:40 PM in concert with
a doubling of fluctuation amplitude. As the other two IRGAs and HMPs continued to indicate increasing air humidity,
$\Delta\rho_{v,LI-7500RS}$ steadily rose for nearly one hour until $\rho_{v,LI-7500RS}$ again agreed with the other instruments. Because only the
averaging period between 9 and 9:30 PM is affected by increased variance water vapor, the spectrum corresponding to that
half-hour is the period with a shift towards higher frequencies.

Cospectra were viewed through the lens of atmospheric stability because it predicts their shape according to Monin-
Obukhov similarity theory (Kaimal and Finnigan, 1994). For all cospectra, the LI-7500 tends to have greater energy in the
production and dissipation spectral regions while being nearly identical in the inertial subrange, and these differences translate
into higher latent heat fluxes (Fig. 6). Lower frequency components of flux were clearly greater, especially in unstable and
neutral conditions, as observed by the LI-7500 (the oldest version), compared to the LI-7500A and LI-7500RS. While the two
newer sensors exhibited similar behavior and relatively smaller fluxes than the LI-7500, under unstable conditions the LI-
7500RS showed a difference in performance from the LI-7500A at high frequencies. For all three IRGA, co-spectra dipped at
2.5 Hz, which should not occur in any desired instruments (Kaimal and Finnigan, 1994). Strong turbulent motions were likely
captured more by the LI-7500A within the surface layer. These cospectra were shifted towards lower frequency compared to
those in neutral and stable conditions, favoring larger eddy sizes with a smaller percentage of energy accumulated in the inertial
subrange (Fig. 6b). This middle frequency range is where the IRGAs were most similar. Regardless of sensor, unstable
conditions featured a flattened peak and more energy towards lower frequencies, as expected for various scalar fluxes measured
with the same instrumentation (Wolf and Laca, 2007). However, in an irrigated cropland environment, the surface layer is
prone to become stable more often than the surrounding area due to a temperature inversion forced by the relatively wetter,
cooler canopy. A previous study demonstrated this effect by using simultaneous sensing over adjacent irrigated cotton and
non-irrigated winter wheat fields, where energy production as depicted by $S_{\rho_v}$ was two orders of magnitude smaller for the
irrigated field than the non-irrigated field (Prueger et al., 2012). Accordingly, in the present study, variability among cospectra
was small under these conditions with relatively few large eddies (Fig. 6e). In contrast, under neutral and unstable conditions,
the LI-7500 departed largely from the other two sensors with energy contribution from low frequency eddies.



### 3.3 Water vapor fluxes

For much of the study period, $\lambda E$ from the LI-7500RS and LI-7500A were similar with slightly larger magnitude than the LI-7500. Overall interinstrument variability $CV_{I-I}$ of $\lambda E$ was 20%, about that of the underlying water vapor variance, and errors on average were less during daytime hours than nighttime (Table 2). For an average diel cycle, the largest $CV_{I-I}$ occurred during the middle of the night, rapidly declined after sunrise, reached its smallest value of 10% at 4 PM, and then increased at a relatively slow rate after sunset. On a seasonal basis, there was a slight, nonlinear increase in $CV_{I-I}$ over time, with mean values increasing from approximately 16% to 24%. Overall, the LI-7500 measured a 15% greater flux than the LI-7500RS both on average and during only daytime hours. Meanwhile, LI-7500A and LI-7500RS fluxes were nearly identical, with 0.5% less flux measured by the LI-7500A and an additional 0.2% difference during the daytime. While the daily bias was as equally positive as negative, the LI-7500A tended to underestimate flux through the first and last third of the study period although possible rainfall effects exist. Greater flux was observed on 27 of the 41 days from DOY 196 – 226, which coincided with greater accumulated ET (Fig. 7). Relative error varied little by time of day. An increase in variability during advective conditions was due to greater mean (co)variance. Under advective conditions, the coefficient of determination was particularly small (see Table 2), but this coincided with large turbulent fluxes including downward sensible heat that was also slightly biased towards increased magnitude.

The 90–day ET (Fig. 7) was in good agreement among the three IRGAs, with slightly greater seasonal flux from the LI-7500, consistent with the larger variance in the timeseries of $\rho_v$. Systematic underestimation of ET for all IRGAs is consistent with advective conditions, especially in the earlier part of the growing season where the gap in daily ET is particularly large for a similar magnitude of ET (Fig. 7). Even if all spectral loss is corrected for, based on the conservation of water vapor and eddy covariance theory, the measured EC flux should be less than the true flux under advective conditions. Approximately 16% of accumulated ET was underestimated from LI-7500A or LI-7500RS relative to the accumulated lysimeter ET at the end of the growing season (Fig. 7). However, only less than 5% of accumulated ET was underestimated from the oldest LI-7000 analyzer (Fig. 7). Furthermore, the EC and lysimeter should differ more with increasing mean ET because the advective component of ET, not captured by EC systems, is more likely to be elevated (Alfieri et al., 2012).

The greater flux from the LI-7500 occurs nearly symmetrically on a diel basis, with relative differences smallest during the day. The mean daytime error of measured flux $\lambda E$ between the LI-7500A and LI-7500RS systems was 4.5%, with the LI-7500A estimating greater ET than the LI-7500RS on approximately three out of every four days. Systematic error $\delta$ averaged 0.08 mm for the LI-7500RS system, which is rather large considering the mean measuring flux of 0.2 mm. Larger systematic error is typically associated with greater flux underestimation due to failure to capture all low frequency signals, consistent with the observed cospectra (Vickers and Mahrt, 1997). In contrast, daily $\lambda E$ differed by 18.6% between LI-7500 and LI-7500RS systems and the magnitude from the LI-7500RS only exceeded that of the LI-7500 on a single day. Comparing daily ET as a function of error, systematic error $\delta$ calculated as shown in Eq. (2), decreases during the study period consistent with



declining ET (Fig. 8). Random error ε was overwhelmingly similar among the sensors, indicating that uncertainty due to sampling has little effect on differences in estimated ET.

## 4 Discussion

### 4.1 Water vapor variance errors

Water vapor variance and flux were compared from three similar eddy covariance systems yielding similar results in rain-free periods. Large $\overline{\rho_v}$ errors occurred under relatively small flux conditions, primarily with the LI-7500A systems. A pattern of increasing flux error corresponding with greater water vapor density error as observed by Fratini et al. (2014) was not found. This is encouraging despite demonstrated substantial errors in the water vapor density measurements. More important for flux is the (co)variance of the water vapor density. Despite screening the data for quality, several outliers were observed in the $\overline{\rho_v'^2}$ which contributed to deflated $r^2$ values and notable discrepancies in water vapor fluxes. Overestimation of water vapor variance could contribute to overestimated flux but is not necessarily the case (Mauder et al., 2006). At noon on DOY 190, LI-7500A overestimated $\overline{\rho_v'^2}$ by 5.9 $g^2$ $m^{-6}$; while corresponding values were only 0.63 and 0.11 for LI-7500 and LI-7500RS, respectively. The overestimation was accompanied by an uptick in flux of 180 W $m^{-2}$, whereas values were 138 and 88 for LI-7500 and LI-7500RS, respectively. In contrast, for the half-hour beginning at 3:00 PM LST on DOY 181, LI-7500 underestimated flux by approximately 20 W $m^{-2}$ despite an overestimation of $\overline{\rho_v'^2}$ (3.5 $g^2$ $m^{-6}$ and 0.84 $g^2$ $m^{-6}$ greater relative to LI-7500A and LI-7500RS). However, in a vast majority of cases, large $\overline{\rho_v'^2}$ was observed with both the LI-7500 and LI-7500A relative to the LI-7500RS and were associated with a recent rainfall event. For instance, a large discrepancy in $\overline{\rho_v'^2}$ among the three IRGAs occurred an hour after light rain on DOY 211, which suggests that thick water droplets may have been still evaporating from the mirror surface. Antecedent conditions were dry and with the cessation of precipitation, a sudden increase in mean wind speed from under 3 to 5 m $s^{-1}$ and a wind shift from east to south enabled sensible heat advection as clouds began to dissipate. Although air humidity decreased by the end of the half-hour for all IRGAs, the magnitude measured by the LI-7500 was much smaller at the start of the averaging period than at the end, in contrast to observations by the LI-7500A and LI-7500RS. Further, we observed that the LI-7500A air humidity began decreasing within the first 15 minutes, suddenly increased by approximately 5 g $m^{-3}$, and then began a rapid decrease. This pattern is different than what was observed by the LI-7500RS, which initially increased and then quickly decreased at an earlier time than for the LI-7500A (not shown). Large variability of air humidity in time and space caused large errors of water vapor density. The LI-7500 $\overline{\rho_v}$ decreased to 7.01 g $m^{-3}$ while the LI-7500A $\overline{\rho_v}$ increased to 17.92 g $m^{-3}$. These corresponded to $\Delta\rho_v$ of 8.83 g $m^{-3}$ and 1.95 g $m^{-3}$, respectively. While the LI-7500RS performance during this time was markedly better than that of the other two sensors, the -1.13 g $m^{-3}$ bias was still different from its long-term offset. Resulting water vapor fluxes were smallest for the LI-7500A and largest for the LI-7500, with sampling by the LI-7500RS seeming to best reflect the variations in eddies during a period of substantial air mass change. A similar event occurred on DOY 196. However, for the half-hour of interest, a relatively small difference in $\overline{\rho_v'^2}$ of the LI-7500





and LI-7500A resulted in a larger flux difference, in which a large, likely overestimated flux was measured by the LI-7500. Interestingly, 20 Hz fluctuations for all systems were dampened during roughly the first half of this averaging period, showing

signs of low frequency atmospheric motion. Once turbulence became more typical of a well-mixed boundary layer, the amplitude of $\rho'_v$ then grew with a larger variance noted in the LI-7500A and LI-7500 compared to the LI-7500RS. This is exactly what was observed on DOY 211 during its relevant averaging period. The effect of rainfall may linger depending on its timing. All the sensors exhibited some degree of non-stationarity in the $\rho_v$ timeseries from late night on DOY 224 into the early morning of DOY 225. However, only the LI-7500A continued to exhibit this behavior for several more hours while the

other two sensors showed constant flux. Because $\rho_v$ was so similar between the LI-7500 and LI-7500RS, it seems that the LI-7500A was uniquely sensitive to the intermittent turbulence during this calm period. Based on the combination of high relative humidity and light winds, these observations were subject to increased random error as expected.

**4.2 Water vapor flux errors**

In the context of ET measurement, total daily magnitude is of prime importance for practical applications. Therefore, flux

errors during the daytime, roughly between 09:00 and 17:00 LST, contribute to the vast majority of ET variation. The similarity between the LI-7500A and LI-7500RS fluxes is reflected by the lack of scatter in covariance data. As expected, errors were larger during advective periods than for other times, but overall correlation between $\overline{\rho'^2_v}$ and $\lambda E$ errors was weak. Highly advective conditions have been associated with large interinstrument variability (Alfieri et al., 2011).

Uncorrected fluxes were assessed to assure that the data processing steps did not appreciably affect our findings. Post-

processing of turbulent fluxes could increment fluxes while causing greater error (Irmak et al., 2014). The magnitudes of $a$, $d$, and $rmsd$ were slightly smaller for all comparisons, and $b$ and $r^2$ were nearly identical, indicating that the corrections contributed little to measurement uncertainty. For instance, the $rmsd$ decreased by 6.8% and 7.3% for daytime fluxes against the LI-7500 and LI-7500A, respectively. Among the corrections, sensor separation and frequency response were of most interest for the LI-7500RS and LI-7500A pair since they are newer different optical analyzers. This may be why among the

three generations of IRGAs, the LI-7500RS consistently had a larger spectral correction factor by approximately 2 to 4%, but again, this served to only slightly decrease flux error. Its midday mean value of 1.11, though slightly larger than for the LI-7500 and LI7500A, was still less than reported in a feedlot for an LI-7500 and CSAT-3 EC system (Prajapati and Santos, 2017). This suggests that high frequency attenuation was relatively minor when turbulent intensity was large, and any missing flux was more attributable to low frequency. While the LI-7500 high frequency energy compared more favorably to the LI-

7500A than the LI-7500RS, a large departure from the LI-7500A and LI-7500RS pattern was clearly observed at low frequencies (Fig. 6).

It has previously been shown that turbulent flux error partitions into primarily random error, with daytime systematic error only as large as 0.018 mm (30 min$^{-1}$) (Alfieri et al., 2011). In contrast, Sect. 3.3 demonstrated that the magnitudes of systematic error were generally large in response to daytime energy balance residuals. The different findings are based on different





assumptions of what is true latent heat. In the prior study, the mean of multiple EC measurements was considered the true flux, and the systematic error was the variance of residuals between predicted and true flux. Following that approach, daytime error was comparable and ranged from 0.014 (LI-7500RS) to 0.024 (LI-7500) mm (30 min$^{-1}$). Also, the prior study was conducted during the period of rapid LAI increase of a cotton crop, while the present study was performed during both the period of rapid LAI increase and crop maturation.

## 5 Conclusion

The guidelines written by Fratini et al. (2014) can be used to avoid water vapor concentration errors. Even in the event that absorptances are not output via datalogger code, and detection of contamination in real time is not done, the water vapor concentration errors will not adversely affect accuracy of eddy covariance on a growing season timescale. Instead, larger fluxes were found from the older LI-7500 system. Our study indicates that the LI-7500 outperformed newer LI7500A and 7500RS

sensors in terms of accumulated ET comparison with lysimeter observations. While it was paired with a different sonic anemometer than the other two IRGAs, flux differences were attributed to differences in variance of turbulent fluctuations of water vapor rather than sonic anemometer error.

Differences in the response from the same model sensor measuring presumably the same air parcel were identified. In this study, the growth and maturation of corn crop drove a change in turbulent flux partitioning. Increases in interinstrument

variation for both water vapor variance and flux were observed when conditions were advective during the period of peak canopy development. Following precipitation, while performance characteristics were consistent in well-mixed turbulent air, larger interinstrument variation was observed under light winds that could cause variation in effects on the IRGA. Adjusting measured fluxes by the systematic error, which tended to be larger at one EC tower compared to the other, brought the water vapor fluxes into strong agreement.

**Data availability**

The associated data are available upon request.

**Author contribution**

SK conducted experiment, collected data, data analysis, and write the manuscript. XL and SK conceptualized the research and developed the methodology. XL and RA were responding for funding acquisition, project supervision, and writing the

manuscript. All USDA authors were responding for helping calibrations, installation, and site maintenance. LX and CO provided infrared analysers and sonic anemometers. All authors reviewed and edited the manuscript throughout the publication process.

**Competing interests**



The authors declare that they have no conflict of interest.

## Acknowledgements

We thank the collaborative scientists and staff in Bushland for their courtesy in allowing access to the experimental site. This work was supported by the Ogallala Aquifer Program which is funded by a USDA ARS research initiative (USDA-ARS 58-3090-5-009), as well as the National Institute of Food and Agriculture under award number 2016-68007-25066.

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



**Figures and Tables**

**Table 1: Performance characteristics of LI-7500A and LI-7500 with reference to the LI-7500RS for water vapor variance $\overline{\rho_v'^2}$. These include regression offset value (a), regression slope (b), coefficient of determination ($r^2$), mean absolute bias (d), and comparability (rmsd).**

| | $\overline{\rho_v'^2}$ | a (g² m⁻⁶) | b (−) | r² | d (g² m⁻⁶) | rmsd (g² m⁻⁶) |
|---|---|---|---|---|---|---|
| | All | 0.04 | 1.17 | 0.42 | 0.09 | 0.88 |
| 75 | Daytime | 0.06 | 1.08 | 0.49 | 0.10 | 0.75 |
| | Advective | 0.22 | 1.02 | 0.57 | 0.24 | 1.33 |
| | All | 0.02 | 1.07 | 0.56 | 0.04 | 0.60 |
| 75A | Daytime | 0.02 | 1.03 | 0.79 | 0.03 | 0.36 |
| | Advective | 0.08 | 1.02 | 0.83 | 0.09 | 0.63 |


**Table 2: Performance characteristics of LI-7500A and LI-7500 with reference to the LI-7500RS for corrected latent heat fluxes (λE). These include regression offset value (a), regression slope (b), coefficient of determination ($r^2$), mean absolute bias (d), and comparability (rmsd).**


| λE | | a (W m⁻²) | b (−) | r² | d (W m⁻²) | rmsd (W m⁻²) |
|---|---|---|---|---|---|---|
| | All | 6.20 | 1.12 | 0.96 | 27.13 | 54.31 |
| **75** | Daytime | 23.91 | 1.08 | 0.92 | 49.42 | 73.66 |
| | Advective | 47.07 | 1.05 | 0.87 | 69.80 | 97.94 |
| | All | -1.48 | 1.00 | 0.99 | -0.87 | 16.69 |
| **75A** | Daytime | 2.34 | 0.99 | 0.99 | 0.34 | 14.53 |
| | Advective | 2.04 | 1.00 | 0.98 | -0.10 | 21.67 |



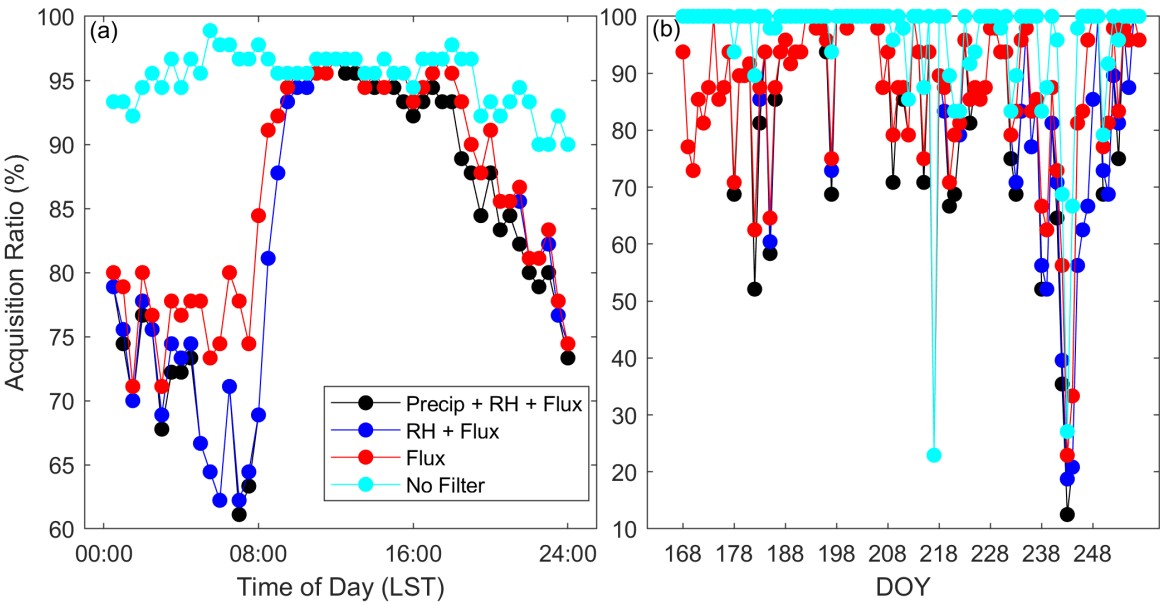

**Figure 1: Screening effects on the data acquisition ratio (AR) as a function of (a) diel cycle and (b) day of year. Precip**
**+ RH + Flux shows AR after all filtering has been completed, RH + Flux indicates AR after RH threshold and steady-state turbulent tests, and Flux denotes AR after only turbulent tests.**

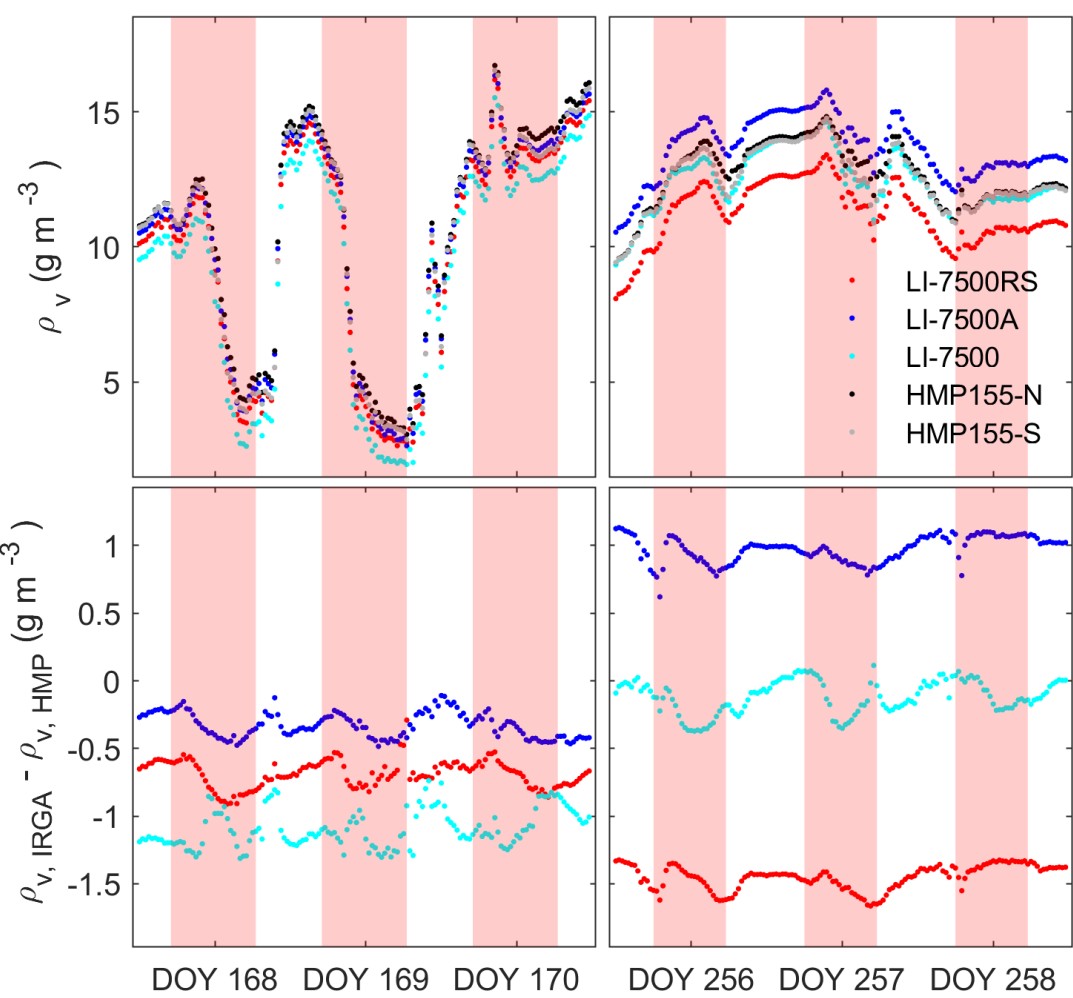

**Figure 2: Absolute humidity ρ$_v$ (top) magnitude and (bottom) difference Δρ$_v$ between paired IRGA and HMP instruments (HMP155-N is paired with the LI-7500RS and LI-7500A; HMP155-S is paired with the LI-7500) during the first and last three days of the study. Shaded areas indicate daytime.**





**Figure 3: Evolution of absolute humidity bias over nine 10-day periods, shown as half-hour bias $\Delta\rho_v$ (points), 1- (thin solid lines) and 10-day (dotted lines) moving averages. Half-hours with observed rainfall are indicated with vertical lines.**


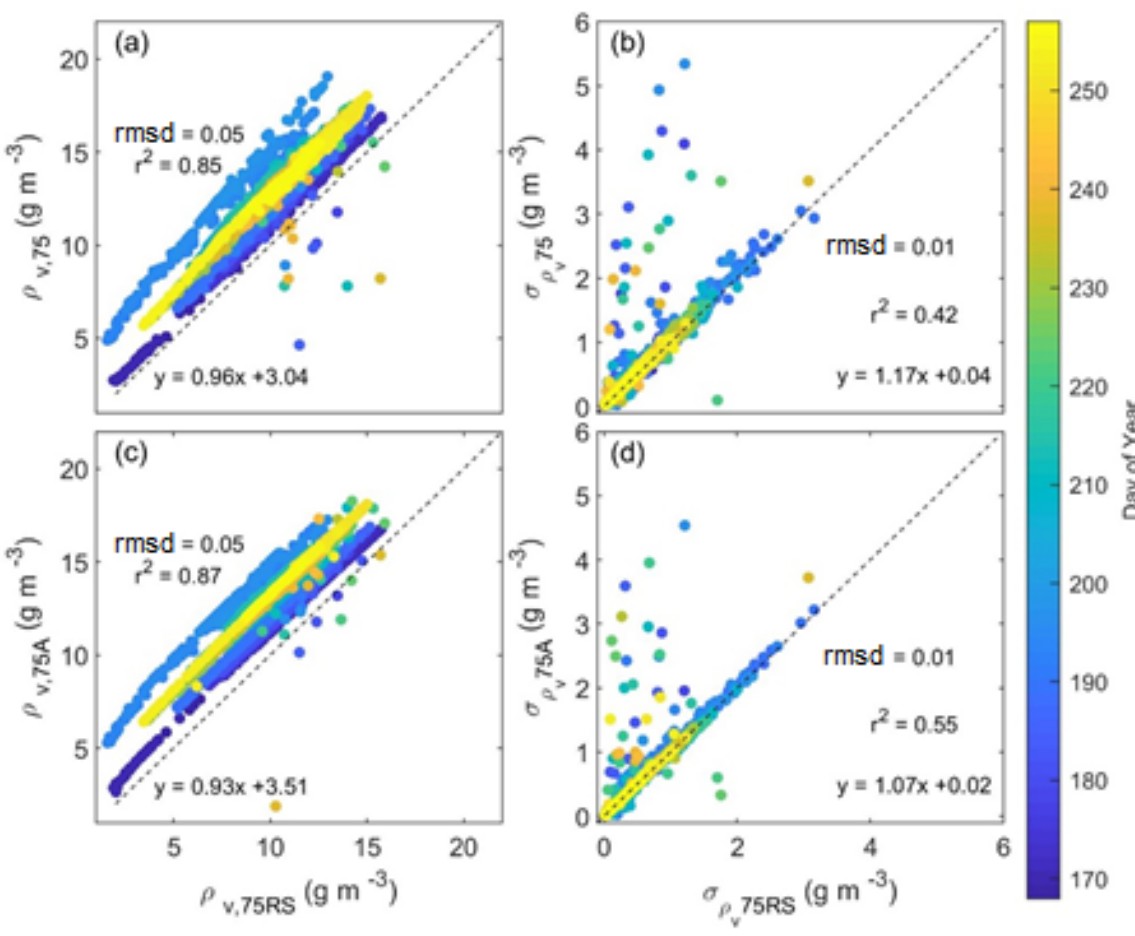

**Figure 4: Intercomparison of absolute humidity $\rho_v$ means and standard deviations for (a, b) the LI-7500 and LI-7500RS and (c, d) the LI-7500A and LI-7500RS.**



**Figure 5: Binned spectra of absolute humidity on DOY 191 are shown for 45 half-hour observations from (a) LI-7500RS, (b) LI-7500A, and (c) LI-7500 as a function of normalized frequency. A close-up comparison of the performance of the three gas analyzers is illustrated in (d) for three half-hours.**




**Figure 6: Ensemble median daytime (a, c, and e) cospectra and corresponding (b, d, and f) ogives under unstable, neutral, and stable conditions. For cospectra, the area between dotted lines shows the interquartile range.**



**Figure 7: Daily ET determined with (a) LI-7500RS (red), (b) LI-7500A (blue), and (c) LI-7500 (cyan). The daily lysimeter ET is displayed by open diamond markers. Accumulated lysimeter ET is shown with solid diamonds and accumulated eddy covariance ET measurements with solid lines.**



**Figure 8: Daytime (9 AM–7 PM LST) ET fluxes for EC systems with an (a) LI-7500RS, (b) LI-7500A, and (c) LI-7500 and the accompanying systematic errors (d–f) and random errors (g–i). Mean values are displayed as larger points.**