# Peer review of "Water vapor density and turbulent fluxes from three generations of infrared gas analyzers"

_Atmospheric Measurement Techniques, 2020_

## Referee Comment (RC1) · Anonymous Referee #1 · 11 Sep 2020

Review of Water vapor density and turbulent fluxes from three generations of infrared gas analyzers, submitted to Atmospheric Measurement Techniques.

This manuscript performs a sensor comparison of water vapor sensors for the eddy covariance method of deriving latent energy fluxes. They compare three version of the Li7500, and find relatively similar (and positive) performance of the three sensors. While generally well put-together and written, there are some gaps. Notably, the pre-publication comments from Referee 2 seem to have been missed and were un-addressed. I know the Copernicus system sometimes makes it difficult to see the attached comments (please find them). Those major comments and mine here below should be addressed and resolved prior to publication.

Important highlights from the missed review that generally denote "major revisions":

1. Define how 20 Hz data spike thresholds of 30+ and 2- g/m3 were determined.

2. The Lasslop et al., 2008 paper seems mis-cited.

3. The use of rmsd is sometimes used to indicate that one instrument is performing well, rather than to indicate merely a difference in performance between instruments.

4. Consider seriously the issues of tranducer shadowing and other non-IRGA instrument errors (including possible errors in Rn, G, and J and why EBC may not be the best metric).

Major comments:

1. "advective conditions" are referred to without a definition of how they were determined.

2. Fig 7 should add the energy balance terms (At least their sum, in accumulation) and then discussed in more detail in the text

3. In general where Fig 7 is described the term "Energy balance closure" should appear at least once (and described, compared to literature, etc.)

4. Discuss the possible differences in CO2 flux (or, if that is coming in a different paper), about the implications of the LE flux differences on the WPL corrections for CO2 or CH4 fluxes (or other gas fluxes).

5. L409 "...no conflict of interest". I fail to see how this can be true. One of the co-authors works for the company that produces these sensors. I don't think it's likely or necessarily an unethical conflict of interest, but it should certainly be stated and justified. The paper helps make the point that this company's sensors are well-suited for purchase and use.

Minor comments:

1. L18 "means" is too jargony. Consider: Water vapor density fluctuation means

exhibited. . ., while their variances were occasionally. . .

2. L19 "following rainfall events" – for how long?

3. L20 "recent" and "results" seem out of place; "widened cospectra" should be quantified.

4. L39 add a paragraph break before "The accuracy of"

5. L46 "optical approaching" seems like the wrong word

6. L52 "showed that zero drift" – sounded like "showed no drift" – "zero" here is a jargon word. Say something more clearly like "drift of the calibration zero, i.e., the bias"

7. L64 define "relatively low"

8. L85 I think both "error" can be "errors"

9. L96 sounds biased; replace "a newer" with "one set of", analyzer with analyzers, and change "earlier" to "other"

10. L100 is there a reference for this field and instrumentation? It's written about as if it is well-known.

11. L114 add "Each" before "gas analyzer" and "the" before "sonic"

12. L155 "rotation" not "rotations"

13. L161 "results of" can be "the"

14. L229 "are" seems jarring after the previous sentences in past tense; I think "Were" is better

15. Fig 1 caption use "turbulence" and not "turbulent"

16. Fig 2 the dots for the HMP155-S are almost impossible to see over the shading.

17. Section 4.1 needs paragraphs; perhaps one with the However in line 333. Consider

also outlining and clarifying the focus and main points; the section wanders.

18. L369 remove "different"

19. L369, 371, 373, and elsewhere – be careful with "This" that lacks a follow-up noun; these words all generate ambiguity.

20. L377 swap "into" and "primarily"

21. L393 add "the" before "corn"

---

## Referee Comment (RC2) · Anonymous Referee #2 · 12 Sep 2020

Review comments on
*Water vapor density and turbulent fluxes from three generations of infrared gas analyzers*
by S Kutikoff, X Lin, SR Evett, P Gowda, D Brauer, J Moorhead, G Marek, P Colaizzi, R Aiken, L Xu, and C Owensby
for **Atmospheric Measurement Technology:** MS no. amt-2020-302.
Date of first review: August 14, 2020; Date of the present (second) review: September 11, 2020.

**Summary Comment**

[Figure]

This manuscript is topical and informative and should be useful to the micromet community. It is certainly appropriate for AMT. My specific comments and recommendation follow.

1    Overall the paper is clear enough, but I still think the use of English could be improved.

2    Lines 133-134 - The authors state "Implausible values of 20 Hz data, defined as greater than 30 g m$^{-3}$ or less than 2 g m$^{-3}$ , were removed . . . ." This range of values poses a bit of a puzzle to me. Why/how were these max/min values chosen and why are they implausible? I think the authors should include a histogram of the noise spikes. They really need to say more about their criteria for noise/spike removal. I will also note that mentioning "Welch's periodogram method" and citing Blanken et al. (2003) (Lines 139-140) does not really address or answer my concern here. Is it possible to show a Welch periodogram and discuss the details relevant to how these "implausible" values were determined?

3    Lines 156-157 - The authors state "Based on spectral losses and other corrections, $E$ was calculated iteratively." This statement needs some clarification. What other corrections are involved and why does $E$ need to be calculated iteratively? It would be helpful to show the equations and explain the need for the iterative approach.

4    Lines 165-166 - Here the authors state "The measured $\lambda E$ is assumed to be the difference between the actual flux and these errors (Lasslop et al. 2008)." This statement also needs some clarification. I do not understand the point of the referring to Lasslop et al. (2008). What exactly does Lasslop et al. (2008) show that is relevant to the authors' study in general and this specific statement in particular? What is the significance of or the need for the Lasslop et al. (2008)

reference. Do Lasslop et al. (2008) state something, either explicitly or implicitly, that is relevant to manuscript that could be restated for clarity?

**5** Lines 166-173 - The definition and discussion of the systematic error must have at least one unstated assumption, i.e., that there are no comparable errors in the heat flux. While this may be true for many eddy covariance systems I don't think one can assume, a priori, that it is universally the case. Could I not define a systematic error (say $\delta_H$) associated with the heat that mimicked Equation (2), i.e., $\delta_H = H(1/ERB - 1)$? If so, what exactly does this mean to the value and utility of using Equation (2) to define the systematic error associated with $\lambda E$?

**6** Lines 215-216 - Here the authors state "After this time, the LI-7500RS appeared to be more stable, with steady $rmsd$ over the final days compared to the other two instruments." This statement also needs some clarification. Because they define $rmsd$ with Equation (1), but this does not seem consistent with their statement. The problem is that they claim that one sensor is more stable than the others, but the $rmsd$ is defined as the difference between two sensors. So how can they claim that the $rmsd$ is a property solely of one instrument?

**7** Lines 390-392 - Here the authors state "While it was paired with a different sonic anemometer than the other two IRGAs, flux differences were attributed to differences in variance of turbulent fluctuations of water vapor rather than sonic anemometer error." At the very least this statement is out of place. It should included in **2.2 Data processing and statistical analysis** or **3.3 Water vapor fluxes** or maybe a separate section devoted to discussing the influence that uncertainties in the other Non-IRGA instruments might have on the present IRGA results. My concern is that there have been at least half a dozen papers in the last 8 years (starting with *Kochendorfer et al.: 2012, Boundary-Layer Meteorology, 145, 383-398* to the most recent *Frank et al.: 2020, Boundary-Layer Meteorology, 175, 203-235*) about sonic transducer shadowing errors causing systematic

underestimation of $w'$. (Note: the other recent sonic papers will be referenced in *Frank et al. 2020*.) So that means the some errors in the water vapor flux that are ascribed solely to the IRGA are in fact caused by the sonic itself. Just how much of an impact does this assumption make on the results of this study? In addition, if $w'$ is biased low, the heat flux, $H$, will also suffer from this bias. So what impact does this have on the $ERB$, Equation (3), and the systematic error $\delta$, defined in Equation (2) and ascribed solely to the $\lambda E$? How certain are the authors that $\delta$ is not dominated by the bias in the sonic vertical velocity rather than errors inherent in the IRGAs? I think the paper would be strengthened if the authors performed a sensitivity or error analysis to estimate how much of $\delta$ is related to non-IRGA errors and how much of $\delta$ can reasonably be ascribed to an IRGA.

**Recommendation**

The paper is acceptably written, but the writing could be improved. I don't think that the statistical analysis is well described. Furthermore, I think the paper approaches this instrument performance problem in a manner that is a bit naive and simplistic. They use the energy balance ratio and its closure as a measure of hygrometer performance. But the measurements of $R_n$, $G$ and $J$ are not free of systematic error or bias. Nor is the sonic necessarily free of bias. How then can they be certain that just because the LI-7500 produces a better closure that it performs better that the other two generations of the instrument? Additionally, they do not discuss possible biases and errors in the lysimeter measurement of ET. I think all sources of errors and uncertainties need to be at least acknowledged in their study. And I think the paper would be further improved it the authors tried to quantify or partition $\delta$ into IRGA and Non-IRGA contributions. Finally, although I would not require a Bayesian statistical approach to their instrument comparison study, I think their efforts and analyses would benefit greatly from such an approach. A Bayesian analysis would allow the authors to build in estimates of the uncertainties associated with the lysimeter and the energy balance instruments.

[Figure]

In turn, this would allow a much more realistic estimate of the inherent uncertainties in the different versions of the LICOR hygrometer and therefore a better estimate of their performance relative to one another. Nonetheless, I recommend the paper for publication after significant revisions.
* * *

---

## Author Comment (AC1) · 14 Oct 2020

Authors used regular fonts for Referee's comments and blue fonts for our responses.

This manuscript performs a sensor comparison of water vapor sensors for the eddy covariance method of deriving latent energy fluxes. They compare three version of the Li7500, and find relatively similar (and positive) performance of the three sensors. While generally well put-together and written, there are some gaps. Notably, the pre-publication comments from Referee 2 seem to have been missed and were un-addressed. I know the Copernicus system sometimes makes it difficult to see the attached comments (please find them). Those major comments and mine here below should be addressed and resolved prior to publication.

[Figure]

Response: Thank you for your review and insight which improved our paper. We responded to all of your comments as well as those from referee #2 .

Important highlights from the missed review that generally denote "major revisions": 1. Define how 20 Hz data spike thresholds of 30+ and 2- g/m3 were determined.

Response: Authors maintain this plausibility range as a way to cover all possible observations in our dataset (they represent the range of growing season values).

2. The Lasslop et al., 2008 paper seems mis-cited.

Response: We removed this reference, as its relation to the error analysis used in this paper does not add clarity to the sentence.

3. The use of rmsd is sometimes used to indicate that one instrument is performing well, rather than to indicate merely a difference in performance between instruments.

Response: We agree on this point. The rmsd is a frequently used measure of the difference between values (sample or population values) predicted by a model or an estimator and the values observed. It is a measure of accuracy in an instrument's performance.

4. Consider seriously the issues of transducer shadowing and other non-IRGA instrument errors (including possible errors in Rn, G, and J and why EBC may not be the best metric).

Response: Transducer shadowing effects were heavily investigated during early 3D sonic anemometer development in the 1980s and 1990s. The optimum design is to minimize shadow effects (air flow distortion dynamics and line/path integration) for the sonic anemometer's geometry (e.g., a 120-degree orthogonal geometry). There aren't many studies on this issue for gas analyzers although strictly speaking, they do have some shadow effects. We agree that there are many non-IRGA instrument errors, especially sonic w component and its spectral property. However, due to surface energy imbalance problems as well as evapotranspiration hysteresis, we considered EBC

as a secondary metric for evaluating the performance of three generations of infrared analyzers.

Major comments:

1. "advective conditions" are referred to without a definition of how they were determined.

Response: Thank-you for pointing this omission out. We have added a brief description and references in our revision as below:

"This condition was defined by finding half-hour observations between 10:00 and 18:00 LST in which latent heat exceeded available energy (a difference between net radiation and soil heat flux), or sensible heat flux was significantly negative ($\leq$ -10 W m$-2$ ) (Kutikoff et al., 2019)."

2. Fig 7 should add the energy balance terms (At least their sum, in accumulation) and then discussed in more detail in the text

Response: Thank you for your insight. At the same site, available energy (Rn-G) and sensible heat flux H (assuming there were no significant differences between the two sonic anemometers we used. One sonic was shared by two IRGAs so that H from them are the same) are the same for three types of IRGAs. The LE (or ET) is of our interest (shown in Figure 7). A composite signal (e.g., energy balance term) might mask the true signal that we are seeking.

3. In general where Fig 7 is described the term "Energy balance closure" should appear at least once (and described, compared to literature, etc.)

Response: Eddy covariance has experienced energy balance closure (EBC) problems over decades. We used the latest version of CSAT3 when we took observations. In addition, two adjacent IRGAs shared one CSAT3 in our study. The EBC problem is not our objective in this study but we recently submitted a paper about energy imbalance problems and evapotranspiration hysteresis to another journal.

4. Discuss the possible differences in CO2 flux (or, if that is coming in a different paper), about the implications of the LE flux differences on the WPL corrections for CO2 or CH4 fluxes (or other gas fluxes).

Response: This is a very interesting topic that is beyond the scope of this paper. We are conducting CO2 spectral analysis from these three analyzers and hope to come out with a different paper.

5. L409 ". . .no conflict of interest". I fail to see how this can be true. One of the coauthors works for the company that produces these sensors. I don't think it's likely or necessarily an unethical conflict of interest, but it should certainly be stated and justified. The paper helps make the point that this company's sensors are well-suited for purchase and use.

Response: Authors conducted this study collaboratively with USDA ARS, Kansas State University, and LICOR-Bioscience. The IRGAs have been widely used in flux communities for nearly 30 years. Our observations and data analysis were objective and unbiased for the purpose of advancing the science. It is not our intention to favor any particular instrument and only present evidence-based scientific results.

Reviewer minor comments 1. L18 "means" is too jargony. Consider: Water vapor density fluctuation means exhibited. . ., while their variances were occasionally. . .

Response: This comment is helpful, and we have made the change as suggested.

2. L19 "following rainfall events" – for how long?

Response: We keep this phrase unchanged because averaged days from one day to a few days are dependent on weather conditions.

3. L20 "recent" and "results" seem out of place; "widened cospectra" should be quantified.

Response: Yes, we deleted "recent" and "results". Thanks for your careful review.

[Figure]

Regarding "widened cospectra", it is not our intent to quantify cospectra specifically but to examine contents of high frequency and/or low frequency energy components. The integral of cospectra is the flux value for the integrating time period.

4. L39 add a paragraph break before "The accuracy of"

Response: Done.

5. L46 "optical approaching" seems like the wrong word

Response: Deleted "approaching".

6. L52 "showed that zero drift" – sounded like "showed no drift" – "zero" here is a jargon word. Say something more clearly like "drift of the calibration zero, i.e., the bias"

Response: Done. Thank you.

7. L64 define "relatively low"

Response: A weekly or bi-weekly calibration is usually required for high accurate measurements from eddy covariance. Therefore, the low instrument stability for an eddy covariance system (fast-response system) is relative to the measurements by a slow-response system, for example, air temperature measurements in a weather station system.

8. L85 I think both "error" can be "errors"

Response: Done.

9. L96 sounds biased; replace "a newer" with "one set of", analyzer with analyzers, and change "earlier" to "other"

Response: Excellent criticism, and we have made these changes.

10. L100 is there a reference for this field and instrumentation? It's written about as if it is well-known.
Response: Added reference to recent publication using this field and instrumentation.

11. L114 add "Each" before "gas analyzer" and "the" before "sonic"

Response: Done.

12. L155 "rotation" not "rotations"

Response: Thank you. Done.

13. L161 "results of" can be "the"

Response: Done.

14. L229 "are" seems jarring after the previous sentences in past tense; I think "Were" is better

Response: Done

15. Fig 1 caption use "turbulence" and not "turbulent"

Response: Done.

16. Fig 2 the dots for the HMP155-S are almost impossible to see over the shading.

Response: We updated this figure to better display HMP155-S.

17. Section 4.1 needs paragraphs; perhaps one with the However in line 333. Consider also outlining and clarifying the focus and main points; the section wanders.

Response: We tightened up, restructured, and condensed the content in this paragraph.

18. L369 remove "different"

Response: Done.

19. L369, 371, 373, and elsewhere – be careful with "This" that lacks a follow-up noun; these words all generate ambiguity.

Response: Great suggestion. Replaced "this" with "this drift" on L212, "this advection coincided" on L298, "These flux results are encouraging" on L324, "This behavior" on L351, and "These findings" on L369.

20. L377 swap "into" and "primarily"

Response: Done.

21. L393 add "the" before "corn"

Response: Done.

— The END of point-by-point response for referee #1

Please also note the supplement to this comment:
https://amt.copernicus.org/preprints/amt-2020-302/amt-2020-302-AC1-
supplement.pdf

---

## Author Comment (AC2) · 14 Oct 2020

Authors used regular fonts for Referee #2 comments and used blue fonts for author's response.

Anonymous Referee #2 This manuscript is topical and informative and should be useful to the micromet community. It is certainly appropriate for AMT. My specific comments and recommendation follow.

1 Overall the paper is clear enough, but I still think the use of English could be improved.

Response: Thank you. Yes, we have made improvements in our revision.

2. Lines 133-134 - The authors state "Implausible values of 20 Hz data, defined as

greater than 30 g m$-3$ or less than 2 g m$-3$ , were removed . . . ." This range of values poses a bit of a puzzle to me. Why/how were these max/min values chosen and why are they implausible? I think the authors should include a histogram of the noise spikes. They really need to say more about their criteria for noise/spike removal. I will also note that mentioning "Welch's periodogram method" and citing Blanken et al. (2003) (Lines 139-140) does not really address or answer my concern here. Is it possible to show a Welch periodogram and discuss the details relevant to how these "implausible" values were determined?

Response: We appreciate you for making this point. It would be worth noting that we used our own Matlab codes to process data and conducted all data analysis including spectral analysis. We did use Eddy-Pro software in this study as well and used to double-check our flux estimates. In fact, water vapor fluxes calculated from both data processing tools were nearly the same. The 2 gH2O m-3 (equals to 111 mmol H2O m-3 or 2.7 mmol mol-1) is the lowest water vapor density during the growing season at Bushland, Texas. The 30 gH2O m-3 is equivalent to 1,666 mmol m-3 or 42 mmol H2O mol-1 water vapor density as an upper bound, which covered any possible highest water vapor density readings in Bushland, Texas. In Eddy-Pro software, the de-spiking thresholds for both water vapor and CO2 are +/- 3.5 standard deviations of a moving window (usually a 5-minute window or 1/6 of flux averaging period with half window overlapped).

We revised the sentence in our revision as: "The data de-spiking process set all data beyond the upper (30 g m-3) and lower (2 g m-3) values as missing. Both upper and lower bounds were estimated by all possible water vapor density observations during the growing seasons in Bushland, Texas."

Regarding Welch's periodogram, it is a method for calculating the power spectral density and co-spectral density in Fourier transform computations. For example, Blanken et al. (2003) used this method for estimating the power spectral density and cospectral density in their 20 Hz time series. This method, per our understanding, is not associated with the upper and lower bounds of water vapor density.

3 Lines 156-157 - The authors state "Based on spectral losses and other corrections, E was calculated iteratively." This statement needs some clarification. What other corrections are involved and why does E need to be calculated iteratively? It would be helpful to show the equations and explain the need for the iterative approach.

Response: Thank you for your insight. Our intent here is to briefly describe the standard flux computation procedures and corrections. We agree that this sentence was not well written and we deleted this sentence to avoid possible confusion.

There are many papers and textbooks that describe iterative approach equations and other standard corrections used in eddy covariance methods (e.g., an excellent software manual by Mauder and Foken, 2004). The basic rationale for having iteration approaches is because the sonic anemometer is directly measuring sonic virtual temperature ($T_s$, $w'T_s'$) rather than absolute thermal temperature ($T_{air}$, for $w'T_{air}'$).

4 Lines 165-166 - Here the authors state "The measured $\lambda E$ is assumed to be the difference between the actual flux and these errors (Lasslop et al. 2008)." This statement also needs some clarification. I do not understand the point of the referring to Lasslop et al. (2008). What exactly does Lasslop et al. (2008) show that is relevant to the authors' study in general and this specific statement in particular? What is the significance of or the need for the Lasslop et al. (2008) C2 AMTD Interactive comment Printer-friendly version Discussion paper reference. Do Lasslop et al. (2008) state something, either explicitly or implicitly, that is relevant to manuscript that could be restated for clarity?

Response: Many thanks for your comments and constructive questions. Lasslop's paper addressed random errors and systematic errors in the eddy covariance system, in which the random errors were estimated by using the gapfilling algorithm (Reichstein et al. 2005). Our objective in this study is to evaluate three generations of IRGAs by inter-comparison, spectral analysis, and direct comparison against an absolute reference – the world-class weighing lysimeter in Bushland, Texas. We also evaluated the systematic errors based on Mauder et al. (2013) and random errors where the estimates were from Finkelstein and Sims (2001). Therefore, we deleted the citation of Lasslop et al. (2008) which is an inaccurate citation in our original manuscript.

5 Lines 166-173 - The definition and discussion of the systematic error must have at least one unstated assumption, i.e., that there are no comparable errors in the heat flux. While this may be true for many eddy covariance systems I don't think one can assume, a priori, that it is universally the case. Could I not define a systematic error (say $\delta H$) associated with the heat that mimicked Equation (2), i.e., $\delta H = H(1/ERB − 1)$? If so, what exactly does this mean to the value and utility of using Equation (2) to define the systematic error associated with $\lambda E$?

Response: This is an excellent point. We agree that our study has to assume that there are no comparable errors in the sensible heat flux. Per our understanding, this is a legitimate assumption. We used two IRGAs to share one cast3 anemometer so that $\delta H = H(1/ERB − 1)$ for the two IRGAs are the same. The second csat3 we used also shared identical homogenous footprints within a well-managed crop field. We tried to examine LE's systematic errors and random errors as our secondary objective in this paper because our main objective was to address intercomparison, spectral analysis, and direct comparison against the weighing lysimeter. We used Eq. (2) to evaluate systematic errors because (1) it can be used to examine the difference between two IRGAs due to insufficient sampling of large-scale air motion; and (2) the EBR in Eq. (2) exactly reflects the energy balance closure problem on a daily basis.

6 Lines 215-216 - Here the authors state "After this time, the LI-7500RS appeared to be more stable, with steady rmsd over the final days compared to the other two instruments." This statement also needs some clarification. Because they define rmsd with Equation (1), but this does not seem consistent with their statement. The problem is that they claim that one sensor is more stable than the others, but the rmsd is defined as the difference between two sensors. So how can they claim that the rmsd is a

property solely of one instrument?

Response: We admit that the rmsd definition by Eq. (1) was not clear for readers in our original manuscript. To clarify, we slightly changed the xRS, i into xREF, i in Eq. (1) and reworded the sentence as below: " please supplement pdf file attached for this equation.

where $x_{(A,i)}$ is the ith observation for the LI-7500/A and $x_{(REF,i)}$ is the ith observation for the reference LI-7500RS. Interinstrument variability was also determined by rmsd except using the average value of three IRGAs or three EC systems as a reference value."

In Figure 3, the rmsd was determined by a reference from the average of three IRGAs water vapor density.

7 Lines 390-392 - Here the authors state "While it was paired with a different sonic anemometer than the other two IRGAs, flux differences were attributed to differences in variance of turbulent fluctuations of water vapor rather than sonic anemometer error." At the very least this statement is out of place. It should included in 2.2 Data processing and statistical analysis or 3.3 Water vapor fluxes or maybe a separate section devoted to discussing the influence that uncertainties in the other Non-IRGA instruments might have on the present IRGA results. My concern is that there have been at least half a dozen papers in the last 8 years (starting with Kochendorfer et al.: 2012, Boundary-Layer Meteorology, 145, 383-398 to the most recent Frank et al.: 2020, Boundary-Layer Meteorology, 175, 203-235) about sonic transducer shadowing errors causing systematic C3 AMTD Interactive comment Printer-friendly version Discussion paper underestimation of w 0 . (Note: the other recent sonic papers will be referenced in Frank et al. 2020.) So that means the some errors in the water vapor flux that are ascribed solely to the IRGA are in fact caused by the sonic itself. Just how much of an impact does this assumption make on the results of this study? In addition, if w 0 is biased low, the heat flux, H, will also suffer from this bias. So what impact does

this have on the ERB, Equation (3), and the systematic error $\delta$, defined in Equation (2) and ascribed solely to the $\lambda$E? How certain are the authors that $\delta$ is not dominated by the bias in the sonic vertical velocity rather than errors inherent in the IRGAs? I think the paper would be strengthened if the authors performed a sensitivity or error analysis to estimate how much of $\delta$ is related to non-IRGA errors and how much of $\delta$ can reasonably be ascribed to an IRGA.

Response: Thank you for these insightful comments. We deleted the "While it was . . ." statement because it was out of place. In sections 2.2 or 3.3 we had similar statements.

We agree that the sonic anemometer's w0 underestimates (vertical component) have been (re)examined in many papers. In 2012 and 2013, two co-authors in this paper intensively discussed shadow effects with some of the authors that you mentioned. We also agree with your insight in terms of sonic uncertainties. However, such uncertainties as well as non-IRGA errors are not the objective for this paper. Our purpose was to address water vapor density measurements and corresponding flux estimates (i.e., latent heat flux) from three generations of IRGAs.

Recommendation The paper is acceptably written, but the writing could be improved. I don't think that the statistical analysis is well described. Furthermore, I think the paper approaches this instrument performance problem in a manner that is a bit naive and simplistic. They use the energy balance ratio and its closure as a measure of hygrometer performance. But the measurements of Rn, G and J are not free of systematic error or bias. Nor is the sonic necessarily free of bias. How then can they be certain that just because the LI-7500 produces a better closure that it performs better that the other two generations of the instrument? Additionally, they do not discuss possible biases and errors in the lysimeter measurement of ET. I think all sources of errors and uncertainties need to be at least acknowledged in their study. And I think the paper would be further improved it the authors tried to quantify or partition $\delta$ into IRGA and Non-IRGA contributions. Finally, although I would not require a Bayesian statistical approach to their instrument comparison study, I think their efforts and analyses would

benefit greatly from such an approach. A Bayesian analysis would allow the authors to build in estimates of the uncertainties associated with the lysimeter and the energy balance instruments.

Response: Thank you for your nice review and insightful comments which substantially improved our paper's quality. Our main objective was to address three generations of infrared analyzers with respect to water vapor density and water vapor flux by using intercomparison, spectral/co-spectral analysis, and direct comparison with the weighing lysimeter. The statistical method we used for systematic errors and random errors was a complementary method in our study. The sonic's uncertainties and non-IRGA errors are beyond for the scope of this paper. It would be our goal to further investigate these uncertainties in the near future including Bayesian analysis.

— The END of point-by-point response for referee #2

Please also note the supplement to this comment:
https://amt.copernicus.org/preprints/amt-2020-302/amt-2020-302-AC2-supplement.pdf

---

## Author Response (AR2)

Point-by-point Responses to Anonymous Referee #1 for #amt-2020-302 R3

Authors used regular fonts for Referee #1 comments and used blue fonts for author's response and red fonts for changes in our R3 revision. We followed referee's all comments and suggestions. Authors appreciate Editor and the Referee that allowed us to improve our revision.

Anonymous Referee #1

Major comments :

1. The authors ignore the suggestion to discuss transducer shadow effect (which could be introduced in the paragraph L100, and elsewhere). Doing so will heighten a sense of the work's relevance and move it from an instrument comparison to a flux comparison. Take some of the response from L967+ and put it into the text.

**Response:** Thank you. Accordingly we moved our R2 point-by-point response to the revised manuscript around L100 as below:

"It should be noted that transducer shadowing effects were heavily investigated during early sonic anemometer development in the 1980s and 1990s. The optimum geometry design minimizes shadowing effects (air flow distortion dynamics and line/path integration) for the sonic anemometer's geometry (e.g., a 120-degree orthogonal geometry) but this …"

2. The authors should comment on potential impact on CO2 flux via the WPL correction if not the CO2 flux differences themselves. This is an important implication of this work

**Response:** Please allow us to explain the reason that we haven't discussed $H_2O$ WPL correction in our second revision (R2). This is because WPL correction for $H_2O$ flux is dependent upon covariance components (i.e., $\overline{w'\rho'_v}$ and $\overline{w'T'}$) but not upon the mean component of $\overline{\rho_v}$ and $\overline{T}$. The $\overline{w'\rho'_v}$ was well explained by the co-spectral analysis shown in Fig. 6. Also the $\overline{w'T'}$ was calculated from sonic CSAT3 that is independent from IRGAs. Both $\overline{\rho_v}$ and $\overline{T}$ in the WPL correction are taken from the slow sensor HMP155. The bias or the drift of $H_2O$ density from the fast sensor (IRGA) doesn't affect the water vapor flux. The HMP155 (or early version HMP45) sensors are stable and sufficiently accurate for mean water vapor density and mean air temperature. The WPL corrections [see below two equations for $H_2O$ flux ($F_v$) and $CO_2$ flux ($F_c$) ] were coded in our site's datalogger or eddy covariance software tool that we developed or Eddy-Pro tool by LI-COR. Two equations for WPL corrections for $H_2O$ and $CO_2$ flux in open-path system are:

$$F_v = \left(1 + \mu\sigma\right) \cdot \left( \overline{w'\rho'_v} + \overline{\rho_v} \cdot \frac{\overline{w'T'}}{\overline{T}} \right)$$

$$F_c = \overline{w'\rho_c'} + \mu \cdot \frac{\overline{\rho_c}}{\overline{\rho_a}} \cdot \overline{w'\rho_v'} + (1+\mu\sigma) \cdot \overline{\rho_c} \cdot \frac{\overline{w'T'}}{\overline{T}}$$

Unlike $H_2O$ WPL corrections, therefore, $CO_2$ WPL corrections additionally require stable and accurate mean terms $\overline{\rho_c}$ (note all $\overline{T}$, $\overline{\rho_v}$, and $\overline{\rho_a}$ are determined by slow sensor HMP155). This is why high-quality $CO_2$ flux monitoring usually requires a weekly or biweekly on-site $CO_2$ calibration due to $CO_2$ density drifts (or offsets).

Therefore, authors agreed to comment on $CO_2$ WPL corrections in our R3 manuscript. We added below at the beginning of subsection '**4.2 Water vapor flux error**' around L470 as:

"The water vapor flux is generally not affected by three IRGA's drifts (or biases) of water vapor density. The WPL corrections for water vapor flux is dependent upon the covariance terms but not upon mean water vapor densities from three IRGAs because the mean water vapor density is determined by the slow response sensors of the HMP155. However, notice that the $CO_2$ flux is certainly affected by IRGA's $CO_2$ drifts because WPL correction for $CO_2$ flux requires the mean $CO_2$ density from IRGA measurements."

3. The authors do not sufficiently highlight their assumptions around rmsd, which are that (1) they implicitly consider the 7500RS as the "good value" and deviation from it as "Bad" even though the article also highlights periods where the 7500RS seems to suffer measurement drift in comparison to the other sensor types (L301). It's better to be clearer that there is no gold-standard and you are assessing differences among sensors and not differences from a standard.

**Response:** We agreed that there is no gold-standard and we used the LI7500RS in figure 4 as a reference for this study's assessment only because it is the newest sensor, not because it was assumed to be the best. In fact, we found that the oldest, LI7500 IRGA performed best for our study.

We modified one sentence to clarify as below in section 2.2:

"In one case, when evaluating sensor drift, the reference is the average of three IRGAs or three EC systems, and in another case, for instrument intercomparison, the reference is the latest sensor, LI7500RS."

4. The reference to upper and lower values is still unclear (L186-8) and feels heuristic or arbitrary.

**Response:** Thank you for your insight. Please allow us to explain this range. They were determined by +/- 3.5 standard deviations in raw data for water vapor density. The 2 g $H_2O$ m$^{-3}$ (equals to 111 mmol $H_2O$ m$^{-3}$ or 2.7 mmol mol$^{-1}$) is the lower bound (-3.5 standard deviations) during the growing season at Bushland, Texas. The 30 g $H_2O$ m$^{-3}$ equivalent to 1,666 mmol m$^{-3}$ or 42 mmol $H_2O$ mol$^{-1}$ water vapor density as an upper bound (+3.5 standard deviations). In Eddy-Pro software, the despiking thresholds for both water vapor and $CO_2$ are +/- 3.5 standard

deviations of a moving window (usually a 5-minute window or 1/6 of flux averaging period with half window overlapped). See Table 1 below:

**Table 1**. Plausibility range for spike detection for each sensitive variable.

| Variable | Plausibility Range |
|---|---|
| *u, v* | window mean ±3.5 st. dev. |
| *w* | window mean ±5.0 st. dev. |
| $CO_2$, $H_2O$ | window mean ±3.5 st. dev. |
| $CH_4$, $N_2O$ | window mean ±8.0 st. dev. |
| Temperatures, Pressures | window mean ±3.5 st. dev. |

(adapted from https://www.licor.com/env/support/EddyPro/topics/despiking-raw-statistical-screening.html).

Authors thus revised the sentence in R3 as,

"Both upper and lower bounds were estimated by using ±3.5 standard deviations of a 5-minute moving window with half window overlapped in water vapor density time series in Bushland, Texas."

5. The description of advection can still be made clearer, and also its effect on the energy balance closure.

**Response:** We agreed. We enhanced the advection description by adding one sentence around L203 which is below:

"Such advected air, usually dry and warm, flowing from adjacent areas to irrigated crop fields is typically the driving force of enhanced daytime latent heat fluxes especially during the afternoon, which may not be fully captured by EC systems and thereby causing reduced energy balance closure."

6. Fig 2 could be improved by increasing the marker size in the legend; the points there are too small for me to detect the difference in blues and greens. (particularly Li-7500 and HMP155S)

**Response:** Thanks for this good suggestion to improve readability. Legend marker size has been increased and the green is now dark orange to improve contrast. The updated figure is shown below:

[Figure]

7. R1 MC2 the energy balance term discussion (L992-7) is not clear and could be still added to Fig 7. It won't confuse the reader to have additional, helpful information there. (see also the point made in response to R1MC3)

**Response:** We agree that additional information would be helpful to link the daily ET to the larger energy balance picture. Here is the energy balance residue from 7500A IRGA.

[Figure]

We added this energy balance residue term in Figure 7. The new Figure 7 is below,

[Figure]

**Figure 7.** Daily ET determined with (a) LI-7500RS (red), (b) LI-7500A (blue), and (c) LI-7500 (cyan). The daily lysimeter ET is displayed by open diamond markers. Accumulated lysimeter ET is shown with solid diamonds, accumulated eddy covariance ET measurements with solid lines. Accumulated daily residual energy is shown in orange circles. Final accumulated energy balance residuals, computed by subtracting eddy covariance ET from the other major energy balance terms ($R_n$-H-G) of each day, for these EC systems (mm): 66.6, 63.3, and 20.0.

8. R1MC5 state explicitly that there is the potential appearance of a conflict of interest but it was managed by the means described in the response (L1020).

**Response:** We believe that is common for scientists at Li-COR, Campbell Scientific, and similar engineering companies to participate in collaborative research.

We inserted a sentence in our acknowledgements:

"It is not our intention to favor any particular instrument and authors only present evidence-based scientific results. There were no financial implications for this study - LI-COR did not fund or incentivize anything related to our findings."

9. R1 minor comment 2, "following rainfall events" – I disagree that this duration should not be quantified. Add "for periods from one to a few days" or something similar.

**Response:** Yes, we agree with you. It certainly can be quantified.  We added "for a period from one to a few days'.

Minor comments :
1. L20 this suggestion was intended to then remove the words "water vapor density fluctuations" from L22, please don't repeat this phrase.

**Response:**  This suggestion makes sense and we have removed the phrase.

2. L309-310 did they see loss in signal strength (RSSI or ADC) during this period?

**Response:** RSSI is for 7700. All 7500 are ADC, in which the first 4 bits represent the signal strength.  Yes, we did see loss in ADC for the LI-7500RS (see below).

[Figure]

3. L375 add apostrophe-s after IRGA

**Response:**  Added s (IRGAs).

4. L445 the transition from general results to one specific half hour seems abrupt. Perhaps adding "for example" and improving the transition would be helpful.

**Response:** Good suggestion. We have connected this sentence better to the previous one.

5. L448 and 449 shift case from singular (was) to plural (were). Pick one and re-write

**Response:** Both are now "were".

6. L467 "exactly" is too precise. Rephrase.

**Response:** Changed to "behavior resembles".

**--- The End of point-by-point response for referee #1**